# The Climate Toolbox—A Microsoft® Excel® Based Tool for Assessing and Comparing the Effects of Internal Climates on Museum Artefacts

**Boris Pretzel** †

Victoria and Albert Museum, London SW7 2RL, UK; boris.pretzel@yahoo.com
† Retired.

**Abstract:** This paper describes the Climate Toolbox—a set of utilities for assessing and comparing real internal climates in terms of hazards they pose to museum collections. The Toolbox is a Microsoft® Excel® workbook with complex VBA codes optimised to process large datasets efficiently and quickly. The Toolbox takes user-provided climate data (up to 8000 sets of temperature, T, and relative humidity, rh, data) and characterises the climates in terms of critical parameters for stresses and mechanical damage, risk and magnitude of mould, relative permanence compared to (selectable) reference specifications, the proportion of data lying within and without selectable specification ranges, and the proportion of rh data swings exceeding a given magnitude. The interface is easily customisable, allowing users to input desired specification ranges, insert opening and closing times (to allow for different temperature specifications for periods when a space is open and occupied to when it is closed and empty), selectively change material critical strain parameters, and adjust the cycle periods for stress analysis. Results are summarised in a range of different graphical and tabular outputs and can be processed further to compare and rank spaces for their suitability to house different collections.

**Keywords:** damage; climate; risk; fluctuations





## 1. Introduction

The decay and degradation of museum and gallery artefacts is inevitable and unstoppable. The rates of degradation, however, can vary over many orders of magnitude and are strongly influenced by the manner in which objects are handled and stored. Failure and loss can occur dramatically and suddenly (acute), over a long term at gradual rates (chronic) or anywhere in between (sporadic). Factors that influence the deterioration and loss of objects have prosaically been termed *agents of deterioration* [1] by colleagues at the Canadian Conservation Institute (CCI) and, somewhat arbitrarily, perhaps out of reverence to Moses and The Commandments, been numbered 10 (Figure 1). (Originally there were nine categories: "physical forces", "criminals", "fire", "water", "pests", "contaminants", "radiation", "incorrect temperature", and "incorrect humidity", with the tenth grouping, "custodial neglect", being added in the mid 1990s [2]). This paper focuses on the evaluation of (internal) climates and the risks different climates pose to museum artefacts, concentrating specifically on deterioration through the agencies of temperature and (relative) humidity. Whilst these are rarely the major factor in acute, sudden loss, the effects of inappropriate climates are unceasing, and examples of objects in which damage can directly be attributed to these factors are ubiquitous. Furthermore, these parameters are, to different extents, within the control of heritage institutions, though their influences on preservation are substantially non-linear and somewhat abstruse.

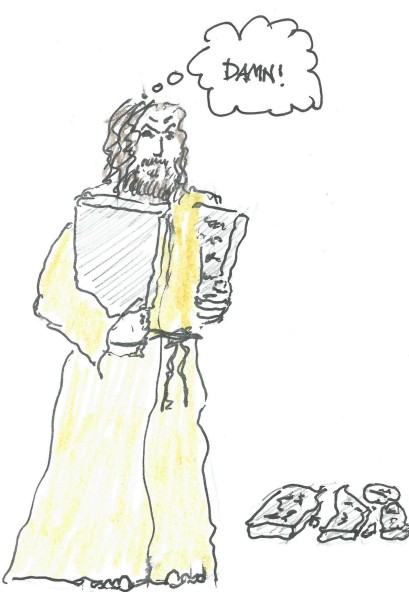

**Figure 1.** The Ten Agents of Deterioration are a framework for categorising causes of degradation and loss in museum and gallery collections. Cartoon ©Nick Pretzel, 2016.

Museums, as places where cultural treasures are housed and displayed, date back certainly to the 15th century (Musei Capitolini, Rome, 1471) though the display of such collections to the public probably only started from the 17th century (Royal Armouries, Tower of London, opened to the public in 1660, and the Amerbach Cabinet, Basel, was in public ownership from the end of 1661, going on public display from 1671). Purpose built museums, as buildings specifically designed for the general public to enjoy art, came along in the 18th century with the opening of the Musei Capitolini building on Capitolini Hill. The 19th and twentieth centuries saw an explosion in museums, with collections ranging from small, local pieces to collections of national and international significance.

Early buildings provided shelter from external environments (rain, wind, direct sunlight) and a degree of heating for visitor comfort in the cold. Nineteenth century buildings, such as those comprising (parts of) the V&A, benefitted from sophisticated ventilation strategies to provide reasonably stable climates and moderate the risk of mould [3]. By the mid-19th century, museums also began to provide artificial illumination (the V&A's Sheepshank Gallery, opened in 1857, was lit by gas lighting and was celebrated in the press as the "the first public gallery ever perfectly lighted by day and gas light", making this the first museum to apply artificial lighting and enabling it to extend public opening well in to the winter evening [4]). By the beginning of the 20th century, evaporative cooling systems became available, hailing the start of the fully air-conditioned building. With developing technologies, and a simplistic view that steady and moderate climates are the most suitable for collections, a virtual arms race started with ever more tightly constrained temperature and relative humidity parameters being demanded for museum collections in the name of better preservation. By the latter quarter of the 20th century, these cycles of increasingly narrow specifications for climate parameters began to be challenged, in terms of their cost effectiveness [5], their achievability [6,7], and their benefits for preservation metrics [8–11].

Irrespective of a museum's (or gallery's) climate specification, a knowledge of the climate is prerequisite for the understanding of its impact on collections and for the design and implementation of any control strategy. The monitoring of internal climates became routine in several London-based museums by the early 1930s. Early records, where they still exist, are in analogue format, making their interpretation very labour intensive. Fur-

thermore, such records are now often quite degraded, adding to the challenge if they are to be made use of. From the 1980s on, a plethora of electronic data loggers became available, simplifying the collection and subsequent analysis of climate data (though many of these electronic devices proved to be unreliable for measuring relative humidity and required careful and frequent calibration). In 1992, the first small scale museum radiotelemetric environmental monitoring system was launched, developed at the V&A in partnership with Meaco (UK) Ltd. [12], further simplifying and extending the routine collection of environmental data. This initial proof of concept system was refined and expanded to cover some stores in the Museum in the mid 1990s, before a new, estate wide, multi-user system (OCEAN—Object-Centred Environmental Monitoring System) was designed and developed (after European wide tender) in partnership with Hanwell Environmental Monitoring from 2002 to 2004 [13,14] (Figure 2). (On completion of the staged installation in 2004, the system comprised approximately 1000 individual units—mainly dual channel temperature and humidity sensors, with a smaller number of lux and UV sensors and a few bespoke sensors for other purposes—making it (one of) the largest ad hoc radio network(s) in Europe at the time).

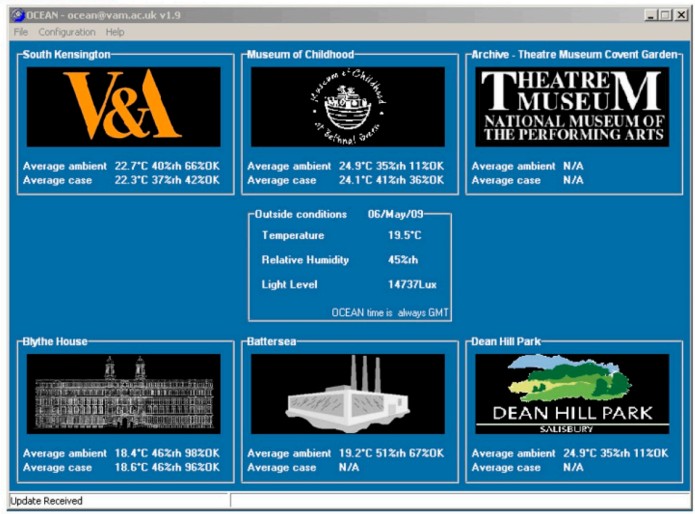

**Figure 2.** The V&A's OCEAN radiotelemetric environmental data monitoring system initial (summary) screen showing current average conditions across a range of sites. (NB the Theatre Museum site was closed at the beginning of 2007 and remained on the system for historical archive data purposes only—no further data were being collected from that site and therefore no live readings are shown). For each location, the current average ambient and (where appropriate) case climate conditions are shown, together with the proportion of sensors reading within their individually assigned specifications. Further detail is obtained when clicking a desired site.

Armed with a comprehensive and growing set of environmental data, the next challenge is to analyse, interpret and use the data to guide decisions for the benefit of the organisation and the collections entrusted to its care.

## 2. Gallery Climates

The Victoria and Albert Museum (V&A) was founded in 1852 and has buildings in South Kensington, London, built between 1857 and the beginning of the 20 Century [3]. Despite the significance of its collection, the air conditioning was introduced to only very few of the Museum's galleries in the course of the 20th century. Resisting the pressure of increasingly tight and difficult (if not impossible) to achieve climate specifications for

collections, the Museum adopted a "Environment Policy" with target ideal parameter in line with specifications elsewhere but much broader "practical" parameters for environments in the galleries. (The Policy was developed in the 1980s and ratified and published by the Museum in 1991 [15]). With only few air-conditioned spaces, the Museum instead relied on the nature of the buildings and the belief that they, together with tightly sealed display cases, effectively buffered internal climates. Practical limits for climate were specified as:

- Temperature always to be within the range of 18–25 °C
- Humidity to be kept within 40–65% rh (with fluctuations of no more than 5% rh an hour).

Analysis of the first full year of data available from OCEAN for the Museums' south Kensington buildings showed that these specifications, broad though they are, were met only about half the time, with low rh, in particular, prevailing throughout the museum in the winter months (Figure 3). Furthermore, the specifications in the (unconditioned) cases in un-air-conditioned galleries were not met for any greater proportions of the time. Even the best air-conditioned galleries in the museum, unless specifically monitored and continuously adjusted, met the practical targets only 71% of the time in this period. Only the exhibition spaces (with a team of engineers on standby and specifically tasked to maintain conditions within parameters acceptable to lenders) performed significantly better. A closer inspection of the climate in different areas showed that some air-conditioned spaces were performing less well than some of the unconditioned spaces and that, with air conditioning strategies in place at the time, there were spaces that were actively being cooled in winter when the external temperature was low (this was probably to help keep the relative humidity levels reasonable but, nevertheless, is a counterintuitive and, probably, counterproductive approach).

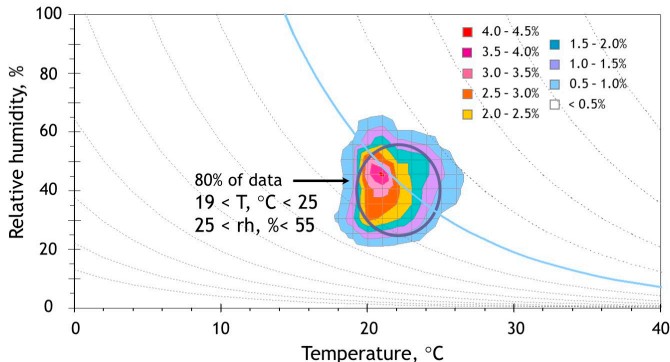

**Figure 3.** Distribution of climate data for all galleries for 1 year (October 2004–October 2005). The surface density plot of the T and rh data is overlaid on a graph showing lines of equal permanence ("isoperms") with the solid blue curve representing the isoperm for reference conditions of 19 °C, 60% rh (see [10,11], Figure 3, for more details). Only 48% of the data fall within the museum's "practical" climate specification, with relative humidity falling below 30% rh for approximately 21% of the time (predominantly in winter). In total, 80% of the data are within the ellipse spanned by temperature form 19–25 °C and rh from 25–55%.

Calculation of the relative proportions of climate readings within a prescribed range is a relatively trivial task. Whilst helpful in identifying when mechanical control equipment is failing to deliver the desired adjustments, and often the basis of conditions of loan, such analysis does not, by itself, yield any greater understanding of the climate or its impact on collections. A common assumption, implicit in the V&A guidelines [15] and many others, is that rapid fluctuation in relative humidity are of significant concern (hence the inclusion of a maximum rh fluctuation of 5% in an hour). The mantra in the mid 1990s was very much along the lines that stable climates with only small variations in rh, are always better than climates with even moderate fluctuations. This assumption was based on pessimistic extrapolation of observed damage in organic material caused by large changes in relative humidity. More realistic considerations and complex models of material responses [8–10]

show that such simplistic assumptions do not predict the likelihood of damage well and more complex analytics are called for.

## 3. The Climate Toolbox

The Climate Toolbox is a standalone set of tools suitable for Excel with complex macros and functions and combined with structured formulae. It is designed for analysing the risks to museum collections posed by varying (internal) climates in a highly efficient and flexible manner. The Toolbox has evolved over the course of the last decade and a half to include more sophisticated and wider ranging risk analyses and allow more user input whilst maintaining at its core highly efficient and optimised routines for the evaluation of large, potentially noisy (i.e., interspersed with spurious values due to signal reception issues) and incomplete (i.e., with missing data due to radio coverage issues or reception interference) climate datasets. On a typical modern desktop, the analysis of over 70,000 point (for example, 1 year's worth of T and rh data collected at 15 min intervals for one location) will take approximately 2 s.

The Toolbox calculations include:

- The proportion of data within given (user selectable) specifications,
- The distribution of the data in selected periods over the T/rh space,
- Risks associated with mechanical damage,
- Relative permanence compared to selectable reference conditions,
- Risk of mould germination and growth.

The Toolbox can also be used to calculate a figure of merit for, or the relative ranking of, different spaces over time in terms of the risk of damage to objects posed by their climates, taking in to account the risk factors computed above, the type and extent of breaches of specifications, and rates of change of climate parameters.

Results are presented in a number of graphical formats, including Climate Footprints [11], Climate Traffic Lights [11], *rhEvent* plots [16], and mould risk assessments (Figures 4–6 in the subsections below). Data are also presented in tabular format (including ranges of temperature and humidity, rhEvents, mould risk, proportion of readings above and below selected temperature ranges—also calculated for occupied periods only, proportion of times above and below selected relative humidities, etc.).

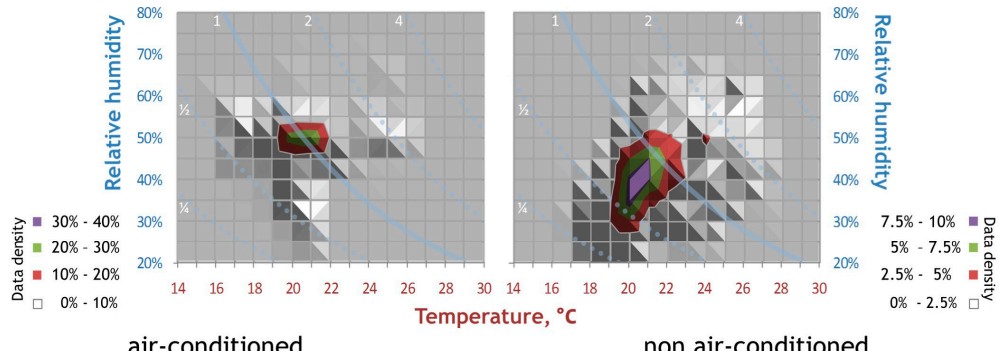

**Figure 4.** Climate footprints for an air-conditioned and a non-air-conditioned gallery in a typical Victorian building. Data from May 2004 to May 2005. The blue curves superimposed over the climate distribution data are isoperms (relative to 19 °C, 60% rh). The air-conditioned space (chart on the left) has data much more tightly clustered, though the two spaces have similar maximum and minimum rh and temperature values, and their effective relative permanence, too, is quite similar.

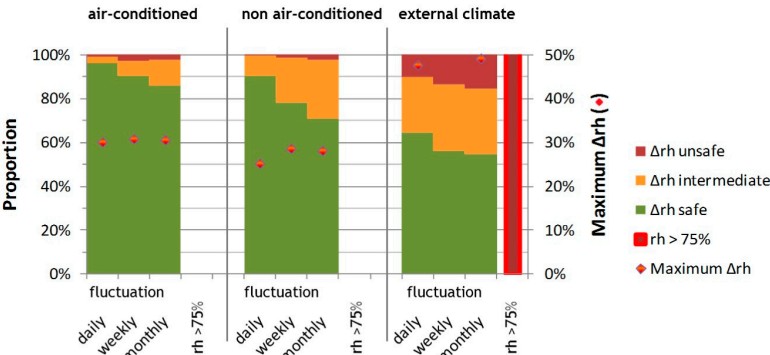

**Figure 5.** Climate traffic lights for an air-conditioned gallery and a non-air-conditioned gallery in a typical Victorian building, together with the external climate. Data from May 2004 to May 2005. Proportions of safe (green), intermediate (amber) and unsafe (red) fluctuations are shown for cycle periods of a day, a week, and a month. The maximum change in rh for each cycle period for each data set is also shown (red lozenge plotted against the right ordinate, i.e., $\Delta rh_{max} \approx 30\%$ for the air-conditioned space for daily, weekly, and monthly cycles. The vertical red bar for the external climate shows that this also had periods of rh over 75%, with associated increased risk of mould and biological attack. (Please note, some object classes, such as corroded glass, minerals containing soluble salts, or archaeological iron, are adversely affected when the rh cycles between specific boundary values. Climate traffic light plots will not represent any reflection of the risk for such objects).

### 3.1. Climate Footprints

Climate footprint plots summarise the distribution of climate data in the temperature—relative humidity plane, preferably with data spanning periods of one (or multiple) years(s) or limited to a single season. The plots do not contain any information on the time or rates of fluctuations but clearly depict the range of the values occupied by the climate. The colours represent different proportions of data at each temperature and humidity (presented with a resolution of 1 °C and 5% rh), with shaded polygons showing changing values in the density. *Isoperms* (curves representing temperature and rh values resulting in constant relative permanence [17]) can be superimposed over the climate footprints and used to judge the effective permanence of the climate data relative to a chosen reference condition [10], as in Figure 4 below. The Figure shows climate footprints of an air-conditioned and an un-air-conditioned gallery in a typical Victorian building. In this example, the climate footprint emphasises the difference in the climates (the air-conditioned space is much more closely clustered, though the maximum and minimum values of T and rh are similar in the two spaces). In the left example, for instance, the green area (20–30% of data) spreads over two temperatures (20 and 21 °C) and a single rh range (50% rh). So, between 40 and 60% of data lie between 20 °C, 50% rh and 21 °C 50% rh. There is also just a spec of purple (30–40% of data) at 21 °C, 50% rh showing that the concentration of data at this point just exceeds a density of 30%. Actual computed values show that 57% of data is concentrated between 20 and 21 °C, 50% rh (26% of data at the lower temperature and just under 31% of data at the higher temperature). Flat grey areas contain no data, whilst the areas containing grey shaded polygons contain a small number of data points each. The isoperms superimposed over the data are computed relative to the reference condition of 19 °C, 60% rh. The two spaces have quite similar relative permanence over the full year, with values close too to that for the reference conditions.

### 3.2. Climate Traffic Lights

Many materials absorb and desorb water with changing rh, undergoing expansion and contraction as they do so. For objects made of such materials, the potential hazards from changing rh and the stresses that can result as structures respond differently to these changes far outweigh thermodynamic considerations. An object's response will depend on the materials from which it is made, the size of (mainly) rh variations to which it is

subjected (for most organic materials, swelling/shrinking on changes in rh far outweigh any changes associated with changing temperature), and the cycle period for the changes. Fast responding materials (thin strips of uncoated wood, parchment, etc.) may respond to changes over a period of a few hours whilst slow responding materials (thicker pieces of wood, coated with water repelling varnish, and the like) will respond fully to changes only over periods of a week or longer. For damage to occur, the change in rh must be large enough to cause sufficient response and occur over a period that is longer than the response time of the materials but shorter that times associated with any stress relaxation mechanisms. Climate traffic light plots summarise the proportions of rh fluctuations (or, in more advanced versions, the stress cycles) that can be considered safe (green = fluctuations under 10% rh), should be treated with caution (amber = fluctuations in the range 10–20% rh), and those with high risk of causing damage (red = fluctuation in excess of 20% rh) over different cycle periods (daily, weekly, monthly, etc.). Such plots are useful for rapid visual comparison of the risk associated with different climates for different groups of objects. The example in Figure 5, below, shows the same data as presented in Figure 4, together with the external climate over the same period (comparison with the external climate shows that the green bars are not very good at conveying risks associated with the climate—even the external climates is green for over half the time. However, large fluctuations in rh, even if not frequent, may cause very significant deterioration and loss. Whilst fluctuations in the green are safe for the vast majority of materials sensitive to fluctuating rh, it is the size of the red and orange bars that show the risk and it may be more appropriate to rescale the plot to give more emphasis to these areas). The plot also shows for each cycle period the largest change in rh (red lozenge, plotted against the right ordinate axis) and a separate red bar for datasets where the rh, on occasion, at least, exceeds 75% (increasing the risk of mould and biological attack). In the current example, rh in excess of 75% did not occur in the galleries.

*3.3. rhEvents*

Climate traffic lights give a good summary for assessing the risks of mechanical damage to a range of artefacts and can easily be adapted to show stresses in materials rather than just magnitudes of rh changes. A starting point for converting the rh changes to stress uses the *domain of allowable rh fluctuation* plots pioneered by Marion Mecklenburg and colleagues [18]. These graphs show the change in rh as a function of equilibrated rh at which different woods exceed their critical strain (≈0.4%) and begin to deform irreversibly, as well as the change to exceed the breaking strain. An example for poplar wood is given in Figure 6 below, showing that, for a piece of poplar fully equilibrated to 30% rh, a reduction to 20% rh will cause stresses exceeding the tensile yield point whilst an increase to 50% rh will cause stresses exceeding the compression yield point.

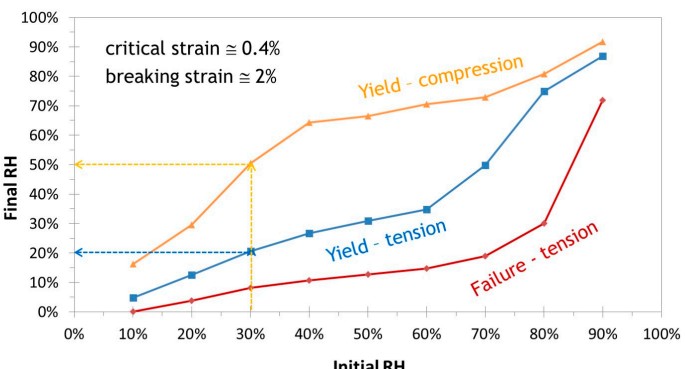

**Figure 6.** Domain of allowable rh variations after Mecklenburg and colleagues ([18]). Data are for poplar, fully constrained in the tangential direction. Initial equilibrium rh is read off the abscissa (in the example illustrated, 30% rh). The blue curve represents the rh for critical strain (0.4%) on shrinkage

(tension—20% rh in the illustration) and the yellow line is for swelling (compression—50% rh in the example). The red curve represents the breaking strain (2%) in tension (8% rh in the illustration). The graph shows that, if a piece of poplar, fully constrained in the tangential direction and fully equilibrated at 30% rh is subjected to a sudden change in rh to 20% (tension) or 50% (compression) and left at the new rh for longer than it takes the wood to respond, it will undergo permanent deformation. (If it is subjected to a change to 8% rh and maintained there, it will crack).

Although climate footprints will show the proportion of the time when safe rh fluctuations are exceeded, they do not allow identification of the events or conditions that have led to the unsafe fluctuations (the temporal data have been removed in the analysis). For building engineers and conservation professionals, however, determining precisely when conditions have tipped from reasonably benign to potentially damaging is important to allow further investigation into possible causes and adaptations to minimise the likelihood of similar changes in the future. This information can be expressed in an rhEvents analysis (Figure 7, [16]). In this approach, the climate data are analysed with multiple (typically two) different time cycles, one representing the (slower) response of a constraining material (e.g., the substrate for layered objects, or the bulk response in a thick wooden object) and one representing the (faster) response of the constrained, more responsive layer (e.g., the top surface in a layered structure or the response of the surface layers of a thick wooden object). As the climate changes, the different response rates will cause build up (and relaxation) of stresses in an object. The slower response rates (longer cycle period) are used to determine the critical rh envelope as a function of time for "safe" changes and the shorter responses are used to determine the extent of compression or tension in the more responsive portions of the objects being considered. Where the lines cross, an event that is likely to cause an irreversible change in the object has occurred. In the Climate Toolbox, the cycle periods and rh response model are user changeable. However, it is sufficient, in practice, to leave the rh response model as set and vary only the cycle periods to predict the extent of mechanical damage that may result in a typical mixed collection of responsive objects (including wooden furniture and carvings, panel paintings, paintings on canvas, ivories, etc.). Suggested typical cycle periods range from a day, week, or month for the short cycle and a month to a quarter for the bulk material, but other values can also be applied for specific object types.

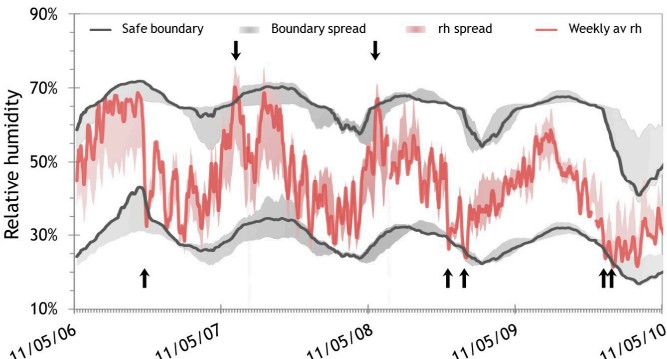

**Figure 7.** *rhEvents* analysis. The Figure shows climate data for a space analysed over two different cycle periods (in this example, a week—168 h and a quarter—2160 h). Solid lines show the data for the space in question, and shaded areas show the spread, taking data from other areas in the dataset into account. Data in red are for the short interval, representing the faster responding areas. Data in black or grey are for the longer cycle, representing the response of the bulkier, restraining material. Where the lines cross, changes in the more responsive layer have exceeded the safe boundary and an "event" with high probability of damage has occurred. In this example, over the 4 years of data represented, seven such events are shown (highlighted by the arrows in Figure 5 in tension—low rh excursions beyond the boundary—and two in compression—high rh excursions. The analysis gives the number of events, their significance (the amount by which critical boundaries are exceeded and

the length of time over which each event persists), and the date and time at which the event occurred. As with climate traffic lights, this analysis will not show risks for objects made of materials containing soluble salts, for example corroded glass, salt laden minerals, or archaeological iron. Dates given as DD/MM/YY.

### 3.4. Mould

Mould occurs in damp conditions at temperatures above 0 °C and is exacerbated by still air. Climate change predictions suggest that the risk of mould in buildings in Europe will increase with future climates. The Climate Toolbox uses two different metrics to calculate the risk of mould associated with indoor climates. The first method uses the approach developed by Sven Thelandersson and Tord Isaksson for what they termed "Mould Resistant Design" (MRD). In their paper [19], the effect of temperature and rh on the relative rate of mould growth is represented by a set of equations and a nominal rate of growth (days to just visible mould with fully viable conidiophores, $D_{crit}$) at give reference conditions (20 °C, 90% rh). The second, more empirical approach uses a look-up table of time to mould germination at different (constant) temperature in the range from 2 to 45 °C and relative humidity in the range from 65–100%, based on observations of mould on wheat, dried beans, and other crops, adapted from data in the Image Permanence Institute's "Dew Point Calculator" [20]. This effectively extends detection of possible mould to conditions where progress is observed to be very slow ( 3 1/4 years at 65% rh, 28 °C).

### 3.4.1. The Mould Resistant Design Model

Thelandersson and Isaksson ([19]) have published equations for the progress of mould germination and growth for temperature in the range 0.1–30 °C and relative humidity above 75% rh. In their model, the equations are used to calculate two daily half doses from the average half daily T and rh readings, and the two half doses are added together to determine the daily dose and progress to germination and visible mould. The cumulative "dose" is proportioned to a selected nominal growth rate (which differs for different materials and boundary substrate conditions). When the ratio exceeds one, visible mould is predicted to be present. For temperatures below 0 degrees or rh under 60%, mould activity is set back from germination at 80% of the rate at which it would grow under the reference condition. So, if the critical reference growth rate is set to 8.5 days (the shortest value given in the paper—appropriate for rough sawn pine—8.5 days to just visible mould at constant 20 degrees and 90% rh, with faster growth at higher temperatures up to 30 degrees) then full die back of activity to dormant spores would need 10 1/2 consecutive days in of such unfavourable conditions. Conditions at temperatures above 0 °C and rh under 75% but over 60% entail a linear setback rate from 0 (no growth, no setback) to 80% of the critical growth rate as rh approaches 60%.

For the Climate Toolbox, the Thelandersson and Isaksson equations have been modified to extend them to higher temperatures than the original model, with an unaltered response in growth rate for temperatures between 30 °C and 35 °C, then a fast drop of the increase in mould activity as temperature increases further—falling off at twice of the rate that activity increases for temperatures from 20 to 30 °C—and ending with an accelerated set back towards pre-germination of 50% per half-day cycle at temperatures above 50 °C (that is, full set back to 0 activity in a day at 50 °C). This high temperature response mimics generally the temperature dependence reported by other authors even if the model is quite broad brush and has not been experimentally verified.

### 3.4.2. Progress to Germination Based on Lookup Table Values

The Image Permanence Institute (IPI) developed a mould risk factor ([20]) of the Climate Notebook and later eClimate Notebook tools. Their model proposes using a lookup table of rates to germination with the daily average temperature and relative humidity and using that to calculate the daily progress to germination. Unfavourable

conditions arrest the cycle to germination but do not set it back. Once germination has been reached or exceeded, unfavourable conditions reset the mould growth back to dormant.

The Climate Toolbox uses a lookup table for the progress to germination but, unlike the IPI model, uses half-day cycles and introduces setback conditions derived from the Thelandersson and Isaksson approach.

Results from the two approaches applied to eight months of data from an example climate (from an unconditioned house basement in the USA) are shown in Figure 8. The results in this example are quite different (not unexpected as they represent different things), though both approaches show significant risk of mould in the space. For this particular example, the lookup model indicates a greater extent of mould germination than the MRD equations but that will not always be the case. Indication of mould activity exceeding 1 in either metric represents a significant risk to collections.

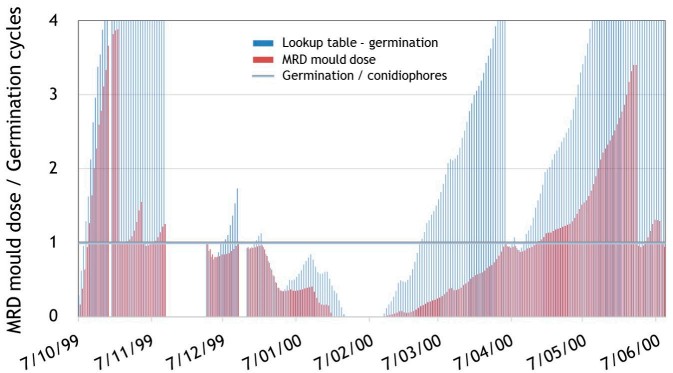

**Figure 8.** Risk of mould. The Figure shows the number of consecutive mould germination cycles (blue bars) based on an enhanced lookup table approach, and the predicted extent of mould activity from the adapted MRD model (red bars). In either case, when the bars exceed the horizontal line at 1, active mould is predicted. The approaches are quite different and show different values, but both predict a high risk of mould for this example data. Dates given as DD/MM/YY.

## 4. Conclusions

The Climate Toolbox is a highly flexible and efficient excel workbook allowing risks to museum objects arising from climates to be evaluated and quantified. Climate data (a date and time column, followed by pairs of columns of temperature and humidity data for each space being considered) can be input at any desired (fixed) collection frequency and for any length of time (within the limits of Excel's row and column restrictions)—though periods of one or more years are preferred for the analyses to take account of any seasonal variations. Base data and cycle periods are selectable by expert users, but default entries can be used for general assessment in the absence of more advanced information on the materials and object properties being considered. Judicious combinations of analyses over different cycle frequencies can be used to determine comparative figures of merits for different spaces and the risks associated with each climate are calculated and presented in terms of a number of risk factors including mechanical damage, relative permanence, risk of mould, overall climate envelope, and adherence to specifications. A copy of the utlity (v 20.0, current at the time of publication), together with supporting files and documentation, is available for download in the Supplementary Material.

The Toolbox is useful for evaluating climate risks for general mixed collections but inevitably will not address all climate related hazards for all materials. In particular, the Toolbox in its current form does not consider hazards associated with fluctuating temperatures (beyond the thermodynamic effect that these will have and the effect on mould germination and growth) so cannot be used to assess risks for collections sensitive to such variations (for example, historic enamel on metal supports). Nor can it adequately assess climate associated risk for materials with embedded salts that can mobilise or recrystallise at specific boundary rh's (such as corroded glass, salt laden minerals, or

archaeological iron). Additionally, as the Toolbox has a minimum cycle period of a day, it cannot assess risks for the very few materials which will respond fully to rh changes in periods shorter than a day. These vulnerable materials all fall outside of the scope considered by the Toolbox and, if present in a collection, will require special attention and control of the local climate.

**Supplementary Materials:** The following supporting information can be downloaded at: https://www.mdpi.com/article/10.3390/heritage6040198/s1, the Climate Toolbox zip file containing a version of the Climate Toolbox, a copy of the User Manual, a license file, the user license agreement, and an example data csv file.

**Funding:** This research received no external funding.

**Data Availability Statement:** Not applicable.

**Acknowledgments:** I am grateful for feedback received from colleagues with whom I have shared earlier versions of the Climate Toolbox.

**Conflicts of Interest:** The author declares no conflict of interest.

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
