# Peer review of "The Climate Toolbox—A Microsoft® Excel® Based Tool for Assessing and Comparing the Effects of Internal Climates on Museum Artefacts"

_heritage, doi:10.3390/heritage6040198_

Round 1

Reviewer 1 Report

I found this an enjoyable article rooted in theory and practice, the tool has clear value in application in the museum sector. I have a few minor revisions to suggest:

The conclusion was quite short and could have elaborated on the benefits of the tool and it's potential impact e.g. how might employing the tool influence decision making around use of space, energy efficiency and sustainability?

The 'safe' fluctuations for different materials and how these are set at 'default' within the tool versus expert user application would be useful to understand.

There were a few areas in the text (see in text comments) where a little additional explanation would be helpful for clarity.

Author Response

Thank you for your detailed and thoughtful review.

I have addressed the highlighted section in the revised draft manuscript. In particular, I have:

  • added more explanation of the climate footprint plot and the meaning of the colours and polygon shapes, and have updated the figure legend to explicitly differentiate between % a proportion of data and %rh
  • have added short warning about materials for which the climate traffic lights and rhevent plots will not represent a reflection of actual risk,
  • have added an extra sentence to explain why emphasis on the size of green areas in the climate traffic lights is less useful in assessing risk to objects than emphasising the size of the red areas
  • have added a sentence to indicate that if either of the mould metrics exceeds 1 then the risk of mould outbreaks will be significant
  • and have expanded the conclusion to also point out further limitations of the toolbox

On the question of the default values versus expert user input, the Tool Box cannot replace the expert user. However, the default values will adequately present risks for the majority of mixed collections including material sensitive to rh fluctuations and by changing the cycle periods under consideration (particularly the short and lone response cycles in the standard analysis), a range of risk scores can be generated to allow reasonable interpretation and evaluation of internal climates that can usefully be used to compare the relative performance of different spaces. For more exacting analysis, the user will need to be aware of the the nature of the response of the most sensitive materials in a collection in a given space to changing rh. In the absence of detailed knowledge of such parameters, the default settings can still provide an empirical score to the climate risk.

In addition to the revised manuscript, I am also submitting a revised support information zip file with an updated version of the ClimateToolBox and the new version of figure 4.

Thank you again.

Reviewer 2 Report

To my opinion, the Climate Tool Box is the very useful tool for assessing the effects of internal climate on museum artefacts.

The benefits of applying this tool are shown. However, I would ask the author to address limitations of this Tool Box.

Author Response

Thank you for your positive comments. I have tried to respond your suggestion of addressing limitations to the Tool box in the extended conclusion and also by highlighting specific classes of materials for which, for instance, the climate traffic light analysis and the rhevents analysis will not adequately represent risk of accelerated damage or deterioration. The tool box also does not temperature fluctuations (other than for the thermodynamic considerations) and therefore cannot, in its current form, be used for assessing climates for materials that are sensitive to temperature fluctuations (such as enamel on silver or copper). Finally, as the Climate Tool Box has been designed to be highly flexible, with many inputs under the users' control, it is possible to enter parameters that are not sensible for a given collection and then extract meaningless or incorrect deductions from the analysis.

I have also updated Figure 4 and take this opportunity to submit as well an updated version of the ClimateToolBox.

Thank you again for your comments.